# Seriniquinones as Therapeutic Leads for Treatment of BRAF and NRAS Mutant Melanomas

**DOI:** 10.3390/molecules26237362

**Published:** 2021-12-04

**Authors:** Amanda S. Hirata, Paula Rezende-Teixeira, João Agostinho Machado-Neto, Paula C. Jimenez, James J. La Clair, William Fenical, Leticia V. Costa-Lotufo

**Affiliations:** 1Department of Pharmacology, Institute of Biomedical Sciences, University of São Paulo, São Paulo 05508-900, SP, Brazil; amanda.hirata@usp.br (A.S.H.); paularez@usp.br (P.R.-T.); jamachadoneto@usp.br (J.A.M.-N.); 2Institute of Marine Science, Federal University of São Paulo, Santos 11070-100, SP, Brazil; pcjimenez@unifesp.br; 3Department of Chemistry and Biochemistry, University of California, La Jolla, San Diego, CA 92093-0358, USA; jlaclair@ucsd.edu; 4Center for Marine Biotechnology and Biomedicine, Scripps Institution of Oceanography, University of California, La Jolla, San Diego, CA 92093-0204, USA; wfenical@ucsd.edu

**Keywords:** natural products, drug discovery, marine pharmacology, antitumor agent, dermcidin, autophagy, apoptosis

## Abstract

Isolated from the marine bacteria *Serinicoccus* sp., seriniquinone (**SQ1**) has been characterized by its selective activity in melanoma cell lines marked by its modulation of human dermcidin and induction of autophagy and apoptosis. While an active lead, the lack of solubility of **SQ1** in both organic and aqueous media has complicated its preclinical evaluation. In response, our team turned its effort to explore analogues with the goal of returning synthetically accessible materials with comparable selectivity and activity. The analogue **SQ2** showed improved solubility and reached a 30–40-fold greater selectivity for melanoma cells. Here, we report a detailed comparison of the activity of **SQ1** and **SQ2** in SK-MEL-28 and SK-MEL-147 cell lines, carrying the top melanoma-associated mutations, BRAF^V600E^ and NRAS^Q61R^, respectively. These studies provide a definitive report on the activity, viability, clonogenicity, dermcidin expression, autophagy, and apoptosis induction following exposure to **SQ1** or **SQ2**. Overall, these studies showed that **SQ1** and **SQ2** demonstrated comparable activity and modulation of dermcidin expression. These studies are further supported through the evaluation of a panel of basal expression of key-genes related to autophagy and apoptosis, providing further insight into the role of these mutations. To explore this rather as a survival or death mechanism, autophagy inhibition sensibilized BRAF mutants to **SQ1** and **SQ2**, whereas the opposite happened to NRAS mutants. These data suggest that the seriniquinones remain active, independently of the melanoma mutation, and suggest the future combination of their application with inhibitors of autophagy to treat BRAF-mutated tumors.

## 1. Introduction

Seriniquinone (**SQ1**, Figure 1A), a natural product isolated from a marine bacterium of the genus *Serinicoccus*, displays potent cytotoxic activity against a panel of tumor cell lines with particular selectivity towards melanoma [1]. This activity is associated with the induction of autophagy and apoptosis [1]. Interestingly, **SQ1** was shown to target dermcidin [1], a small protein for which no small-molecule modulator has been reported. To date, a small panel of **SQ1** analogues, illustrated by **SQ2**–**SQ5**, Figure 1A, have been prepared, displaying an improvement of solubility, cytotoxicity, and modulation of dermcidin expression [2]. Here, we report on the activity of these materials in cell lines representing key mutation profiles observed in melanoma using SK-MEL-28 and SK-MEL-147 cell lines. 

Melanoma, a cancer originated from melanocytes, frequently presents mutations in the MAPK pathway, mainly in BRAF and NRAS proteins, at incidences up to 52% and 28%, respectively, according to The Cancer Genome Atlas Network [3]. This disease has limited therapeutic options and frequent chemoresistance represents a major hindrance [4], as reported for dacarbazine [5], BRAF^V600E^ inhibitors [6], and immunotherapies [7]. Thus, the pursuit for alternatives, with different targets and mechanisms/modes of action (MOAs), is crucial for advancing treatment options for melanoma. Furthermore, the lack of specificity for tumor cells in detriment of normal tissue remains an unmet challenge.

Human dermcidin is a 110 amino acid protein encoded by the dermcidin (*DCD*) gene. This full-length protein is known to be the source of peptides, including the proteolysis-inducing factor (PIF) and antimicrobial peptides constitutively produced by eccrine sweat glands [8]. Despite this protective function, dermcidin has been reported at high expression levels with a suggested survival-promoting activity in select cancers [9], but its role in carcinogenesis and tumor maintenance is still unclear. Further studies have shown that knockdown of *DCD* in breast and lung cancer cell lines repressed cell proliferation and tumorigenesis, therefore reinforcing its relevance for cancer progression [10,11]. Further studies have shown that high dermcidin levels are associated with late-stage melanoma in patients with metastasis, indicating a role in melanoma progression and suggesting a prognostic value as a marker [12]. Here, we further our exploration of the utility of seriniquinone (**SQ1**) and its analogues (**SQ2**–**SQ5**) as agents that target cancer’s bearing point mutations on BRAF and NRAS genes.

## 2. Results

### 2.1. Seriniquinone Analogues Display Cytotoxic Activity in Cancer Cells 

Seriniquinone (**SQ1**, Figure 1A) and its synthetic analogues **SQ2**–**SQ5** (Figure 1A) displayed cytotoxicity against melanoma, breast adenocarcinoma, colorectal carcinoma, and non-tumor cell lines (Figure 1B; Appendix A). The IC_50_ (half-maximal inhibitory concentration) value for **SQ1** in melanoma cells ranged from 0.06 µM in SK-MEL-19 to 1.4 µM for MM200 cell lines, while it reached 3.3 µM in breast adenocarcinoma (MCF7 cell line) and 1.0 µM in colorectal carcinoma (HCT-116 cell line). For the synthetic analogues, cytotoxicity varied from 0.04 to 1.8 µM for melanoma cells and from 0.15 to 1.9 µM for MCF7 and HCT-116. The most prominent effects were obtained for melanoma cell lines (SK-MEL-19, SK-MEL-28, and SK-MEL-147). For fibroblast cells (MRC-5 and FDH cell lines), **SQ1** had an activity of 0.6 and 1.2 µM, respectively. In comparable assays, the synthetic analogues ranged between 0.6 and 3.5 µM for MRC-5 and FDH cells, respectively. Doxorubicin, a positive control, was generally less toxic than seriniquinone and its analogues in melanoma cells, with the exception of WM293A and MM200 cell lines. Nonetheless, it was far more toxic to breast carcinoma (MCF-7) and colon carcinoma (HCT-116) cells, and nontumor cell lines (MRC-5 and FDH).

Selectivity indices (SI, the ratio of the IC_50_ value for the respective tumor cell line to the IC_50_ for MRC-5 or FDH, the non-tumor cell lines) were improved mainly for **SQ2** and **SQ3**, relative to those of **SQ1** (Table 1 and Table 2). SK-MEL cell lines displayed the highest SI values, while **SQ2** presented greater levels of selectivity, reaching a 30–40 ratio favoring melanoma cells (red in Table 1 and Table 2). Doxorubicin showed in the lowest SI values for most cell lines screened (Table 1 and Table 2). To further explore and compare the effects of natural **SQ1** and analogue **SQ2** on melanoma cells, both compounds were subjected to an in depth evaluation of their MOAs using BRAF^V600E^ mutated (SK-MEL-28) and NRAS^Q61R^ mutated (SK-MEL-147) cell lines.

### 2.2. **SQ1** and **SQ2** Are Active in BRAF and NRAS Mutant Melanoma Cells

SK-MEL-28 and SK-MEL-147 cells were treated with **SQ1** or **SQ2** for 24, 48, and 72 h in order to analyze their sensitivity after short (24 h) and long exposure (72 h) periods (Figure 2A). IC_50_ values for these cell lines suggest that **SQ1** and **SQ2** display a time-dependent cytotoxicity, with more potent inhibition of cell proliferation at longer exposures (72 h, Appendix A). Interestingly, compound **SQ2** was more active when compared to **SQ1** in each cell line at 24 and 48 h but showed similar potency after 72-h exposure in SK-MEL-147 cells. In the Trypan blue exclusion test (cell viability), 24-h treatments induced toxicity by significantly reducing the number of viable cells and increasing non-viable ones in either cell lines (Figure 2B). However, for SK-MEL-147, the proportion of non-viable cells was significantly higher when treated with **SQ2** at 2.5 and 5.0 µM.

In order to evaluate the effects of these compounds on long-term survival, cells were treated for 24 h with low concentrations of **SQ1** and **SQ2**, based on their half IC_50_ and IC_50_ values, and followed by clonogenic assay of the remaining viable cells in drug-free medium (Figure 2C). **SQ1** decreased the ability of cells to readhere, expressed by a significant reduction in the number of colonies, and inhibited proliferation, a factor revealed by a lower colony area (Figure 2D,E). Overall, we found that the reduction of the colony levels was greater in SK-MEL-28 (Figure 2E) than SK-MEL-147 (Figure 2D), with reduction levels greater than 50% in both cell lines.

### 2.3. **SQ1** and **SQ2** Modulate Dermcidin Expression

To validate the targeting of dermcidin by **SQ1** [1], RNA transcription of the *DCD* gene was evaluated in treated cells. For SK-MEL-28, **SQ1** modulated *DCD* expression in a dose-dependent manner, increasing almost twofold at 0.3 µM reaching 20 times at 0.6 µM (Figure 2F). In comparison, **SQ2** treatment up-regulated nearly three times only at 0.1 µM. In SK-MEL-147 cells, both compounds modulated *DCD* expression only at the highest tested concentrations (1.8 µM for SQ1 and 1.2 for SQ2), for which **SQ1** reached an expression twofold higher than the control and **SQ2** induced overexpression almost eight times (Figure 2G). These data suggest that structural modifications that yielded **SQ2** did not affect the capability of this seriniquinone analogue to modulate dermcidin; however, it does so with seemingly distinct specificities than those of the natural compound. 

### 2.4. **SQ1** and **SQ2** Lead to Cell Death through Autophagy and Apoptosis

Evaluation of indicators for apoptosis and autophagy revealed that both BRAF and NRAS mutated cell lines are sensitive to **SQ1** and **SQ2**, as evidenced by the increase of typical apoptotic markers, including cleaved PARP-1 and cleaved caspase-3 (Figure 3A). Signs of cellular autophagy were also revealed by decreased p62 or LC3BI-II synthesis. Seriniquinone-treated SK-MEL-147 cells showed more expressive signs of apoptosis. SK-MEL-28 cells treated with **SQ1** had a more prominent response to autophagic flux, expressed through the extinction of p62. Here, protein expression was evaluated in a time-dependent manner following treatment of SK-MEL-28 with **SQ1** (Figure 3B). Significant changes on cleavage of PARP-1 and caspase-3 occurred between 8 and 12 h indicating the initiation of apoptosis, while a concomitant initiation of autophagy was observed through the decrease in the levels of p62 and an increase in LC3B ratios.

### 2.5. BRAF and NRAS Mutants Have Different Basal Profiles

Basal expression of key genes associated with autophagy and apoptosis were quantified by qPCR (Figure 4), due to the different expression profiles of cell death-related proteins between cell lines. Prior to compound exposure, SK-MEL-28 and SK-MEL-147 were found to significantly differed in their expression of 13 of the 18 genes assessed: *BCL2*, *BCL2L1*, *MCL1*, *BAK1*, *BAD*, *BCL2L11*, *BID*, *PMAIP1*, *BBC3*, *BNIP3*, *ATG7*, *MAP1LC3B*, and *XIAP*. Some genes related to apoptosis, *BAK1*, *BAD*, *BID*, *PMAIP1*, *BBC3*, and *BNIP3*, were overexpressed in SK-MEL-147, while *MAP1LC3B*, related to autophagy, was suppressed. Despite a seemingly apoptotic profile in SK-MEL-147 cells, further corroborated by the low levels of *BCL2* and *XIAP*, apoptosis suppressor genes, such as *BCL2L1* and *MCL1*, were overexpressed. Additionally, *DCD* basal expression varied among the independent samples from both lines, however, the overall expression of this gene in SK-MEL-147 was found higher than that in SK-MEL-28.

### 2.6. Inhibition of Autophagy Induces Different Responses in NRAS and BRAF Mutated Cells

Next, we explored the effects of pre-treatment with bafilomycin A1 (an autophagy inhibitor) or rapamycin (an autophagy inducer) to further explore cell death mechanisms elicited by seriniquinones. We complemented this study with a similar approach to understand the role of the seriquinones on apoptosis using pre-treatments with Z-VAD-FMK (an apoptosis inhibitor) or doxorubicin (apoptosis inducer). In both cell lines, it was possible to demonstrate the difference of cell death triggered by **SQ1** and **SQ2** when autophagy is inhibited, expressed by the accumulation of p62 and LC3B. SK-MEL-28 cells were more sensitive to the SQ-treatments, indicated by decrease in cell viability (Figure 5A, Appendix A), corroborating with an increase of apoptosis (Figure 5B). In contrast, SK-MEL-147 increased cell survival following autophagy inhibition, just as the Z-VAD-FMK pan-caspase inhibitor (Figure 5C,D, Appendix A). Also, for both cell lines, pre-exposure to Z-VAD-FMK seem to restore the autophagy signal just as in **SQ1**, **SQ2**, or rapamycin treatment alone. DOX treatments are marked for apoptosis activation and the cell lines displayed the same behavior described for autophagy inhibition through MTT analysis. However, SK-MEL-147 revealed an increase in cleaved PARP-1 and cleaved caspase 3 for this combination, while a decrease in cell death was shown by the results of the MTT assay. Exposure to the pan-caspase inhibitor reduced transformation of 19 kDa cleaved caspase 3 into the 17 kDa fragment and, consequently, less PARP-1 cleavage. Interestingly, p62 reduction and/or LC3B accumulation were observed (Figure 5). IC_50_ values and Western blotting quantifications for this study can be found at Appendix A and Appendix A, respectively.

## 3. Discussion

The main therapeutic barrier for melanoma is tumor aggressiveness and propensity to recur. Both features can be related to development of resistance to available treatments, due, in part, to the malleable metabolism within the tumor microenvironment [13,14]. The high incidence of melanoma mutations occurring in BRAF^V600E^ has prompted a clinical exploration for drugs that selectively inhibit the mutated protein, such as vemurafenib and dabrafenib. However, most patients develop resistant tumors after 5–6 months of treatment [15,16,17]. While RAS has been the first oncogene described for melanoma [18], no successful NRAS mutated protein inhibitor has gained clinical approval. Currently, MEK inhibitors serve as the primary therapeutic strategy for mutated melanoma; however, with modest efficacy [19].

Considering these limited treatments, alternative treatments are necessary. Medicinal chemical studies on the natural product seriniquinone (**SQ1**) have advanced analogues with improved solubility, cytotoxicity, and tumor selectivity [1]. Such efficacy is due to the unique MOA of this compound, which could, indeed, uncover a novel chemotherapeutic target, the dermcidin. Still, little is known on how modulation of dermcidin affects cellular activity and further studies are required. Some of these issues have been addressed through synthesis of analogues of **SQ1**, which displayed higher solubility and similar or improved toxicity [2], representing an important step for the development of a class of dermcidin targeting anticancer agents.

Our studies demonstrated that synthetic analogues (**SQ2**, **SQ3**, **SQ4,** and **SQ5**) maintained or improved the biological activity of the natural compound **SQ1**. Further studies showed that **SQ2**, **SQ3**, **SQ4,** and **SQ5** provided effective selectivity indices (≥twofold) to SK-Mel cells in relation to non-tumor cell lines. Interestingly, **SQ2** showed 30–40-times greater selectivity toward melanoma cell lines. Such selectivity is important to minimize side effects and has been widely explored for the development of novel anticancer drugs [20].

According to Trzoss et al. [1], the cytotoxicity of **SQ1** was time-dependent requiring a 24-h exposure for efficacy in colorectal adenocarcinoma cells (HCT-116). For the BRAF^V600E^ and NRAS^Q61R^ melanoma cells, **SQ1** also displayed a time-dependent activity, increasing its activity between 24 and 72 h. After the initial 24-h exposure, **SQ2** caused a higher decrease in cell viability than **SQ1**. Here, we found that viable SK-MEL-28 cells were able to adhere and proliferate in fresh, drug-free medium, in contrast to those exposed to **SQ1**. For SK-MEL-147, the parameters of colony number and area were also decreased by exposure to either **SQ1** or **SQ2** with cell proliferation, expressed by colony area, decreased only 50%. These data suggest that **SQ1** may produce an irreversible scenario for cell survival in SK-MEL-28. Additionally, these studies provided the first evidence that BRAF^V600E^ and NRAS^Q61R^ melanoma cells may have different responses to **SQ1** and **SQ2**. Supporting this observation, we found that the gene expression of dermcidin, the target of **SQ1**, was modulated at different levels within these cell lines.

Besides the well-established role for the dermcidin in skin defense, this protein has been described as a cancer survival factor [21,22,23,24] with overexpressed in select tumors. Herein, the basal *DCD* expression varied among the biological triplicates from both SK-MEL-28 and SK-MEL-147 cells with enhanced expression in the NRAS mutant cell line. This observation may suggest why SK-MEL-147 cells are more resistant to **SQ1** and **SQ2** treatments during short periods (24 and 48 h) where *DCD* can act as an oncogene. Previous studies have shown that **SQ1** and its analogues modulated *DCD* mRNA expression in HCT-116 (colorectal carcinoma), PC-3M (prostate carcinoma), MALME-3M (melanoma) and SK-MEL-28 (melanoma) cell lines [1,2]. We now find that **SQ1** and **SQ2** also up-regulated the expression of such transcripts in SK-MEL-28 and SK-MEL-147 cells in a dose-dependent manner, suggesting that analogs such as **SQ2** also act by targeting dermcidin.

Autophagy plays an important role in cellular homeostasis, such as regular intracellular maintenance, tissue development and regeneration, aging and cell survival through starvation, and other stress conditions [25]. However, depending on the tumor scenario, therapeutic strategies for cancer treatment involving induction of autophagy is still controversial [26]. Current assessments of autophagy suggest that it acts as a cell survival mechanism, supplying necessary basal conditions for cancer cells to recover and resist transient adversity, which associates with tumor aggressiveness [27]. On the other hand, this cytoprotective autophagy can switch to a cytotoxic phenomenon depending on tumor profile or compound’s MOA, but still through unclear pathways [28,29]. Despite the mutations carried by SK-MEL-28 and SK-MEL-147, both **SQ1** and **SQ2** induced caspase-mediated apoptosis; nonetheless, the cell lines responded differently through autophagy inhibition. The BRAF^V600E^ mutant cells were sensitized by pre-treatment with an autophagy inducer, while the opposite happened to the NRAS^Q61R^ bearing cells. Other groups reported a higher basal autophagy level for BRAF mutant cells, which corroborates with our gene panel data, and abolition of this process also caused a decrease in cell viability due to its relation to the maintenance of functional mitochondria [30,31,32]. However, only recently has the association between NRAS mutant melanomas and autophagy inhibition matured [33,34] despite the existence of a clinical trial in progress exploring the combination of MEK pathway modulators and autophagy inhibitors (NCT03979651). Zheng et al. reported that flunarizine, a drug approved for migraine and epilepsy treatment, triggered cytotoxic autophagy in basal-like breast cancer. When blocked, flunarizine treated cells partially restored NRAS levels [35]. These studies established that that small molecule induction of autophagy could lead to the degradation of NRAS and subsequent processing of RAS proteins in lysosomes [36]. These data support our hypothesis that NRAS^Q61R^ cells also trigger cytotoxic autophagy that, when inhibited, may cause a restoration of NRAS protein levels, which could explain the increase in cell proliferation. Studies are now underway to explore this phenomenon.

Challenges remain in clinical treatment of melanoma. When addressed with the conventional therapy, such as the alkylating agent dacarbazine, melanoma easily turns into a refractory tumor. There are some available options of molecular targeted inhibitors for this type of cancer, like BRAF and MEK inhibitors. However, the fast tumor recurrence is still problematic. Immunotherapies (anti-CTLA-4 and anti-PD-1 immune checkpoint inhibitor) have shown promising outcomes; however, these extensive treatments have shown to be non-responsive in patient cohorts [37]. Consequently, new strategies are urgently needed to manage melanoma. Ideally, the next generation therapies should embrace novel targets to reduce onset of resistance. Therefore, the differential mechanism of activating cytotoxic autophagy, induced by **SQ1** and **SQ****2,** may be key for melanoma treatment [38]. Here, we showed that autophagic dysregulation in melanoma may lead to apoptotic cell death through induction of autophagy in cells that already have a higher basal autophagy activity, as seen in BRAF^V600E^ mutant cells. Furthermore, after triggered, autophagy may turn into a cytotoxic process in cells that have this mechanism suppressed. Here, small molecule modulation provided a response that can be detrimental to their basal maintenance, as evident in NRAS^Q61R^ mutant cells. Further studies are needed to elucidate the relation between dermcidin modulation and autophagy activation. Moreover, additional effort is required to understand the role of dermcidin in tumor cell survival and resistance. Additionally, to advance with seriniquinone into preclinical studies using animal models, it is necessary to overcome its poor water-solubility. As discussed by Apolinario et al. [39], the development of nanotechnology-based formulations, as observed for other lipophilic natural compounds, including paclitaxel and doxorubicin, can be an effective strategy to enable in vivo studies with seriniquinone.

## 4. Materials and Methods

### 4.1. Cell Culture

Human cell lines were used throughout this program. SK-MEL-19 (melanoma BRAF^V600E^, NRAS^WT^), SK-MEL-28 (melanoma BRAF^V600E^, NRAS^WT^), SK-MEL-147 (melanoma BRAF^WT^, NRAS^Q61R^), MRC-5 (fibroblasts), and primary fibroblast (FDH) were kindly provided by Professor Glaucia Santelli (University of São Paulo, Brazil). MCF7 (breast adenocarcinoma, ATCC HTB-22), and HCT-116 (colorectal carcinoma, ATCC 247) cells were obtained from American Type Culture Collection (ATCC) and deposited in the Cell Bank of Rio de Janeiro, Brazil. WM293A (metastatic melanoma BRAF^V600D^, NRAS^WT^), 501mel (metastatic melanoma BRAF^V600E^; NRAS^G12D^), and MM200 (melanoma BRAF^V600E^, NRAS^WT^) were kindly provided by Professor Sharon Prince, University of Cape Town, South Africa. All human melanoma and breast cell lines were cultivated with Dulbecco’s Modified Eagle Medium: Nutrient Mixture F-12 (Thermo Fisher Scientific, San Jose, CA, USA). MM200 and HCT-116 were maintained with Roswell Park Memorial Institute (RPMI) medium (Thermo Fisher Scientific, San Jose, CA, USA). Unless stated otherwise, media was supplemented with 10% fetal bovine serum (FBS) and 1% penicillin-streptomycin (Thermo Fisher Scientific, San Jose, CA, USA). Cells were incubated with 5% CO_2_ at 37 °C. When required, cells were detached from flask using trypsin-EDTA 0.05% (Thermo Fisher Scientific, San Jose, CA, USA).

### 4.2. Seriniquinone Analogues and Control Agents

Unless noted, all compounds were prepared as stocks in DMSO (Synth, Diadema, SP, Brazil). Seriniquinone (**SQ1**) and its synthetic analogues **SQ2, SQ3**, **SQ4,** and **SQ5** (Figure 1A) were obtained as described in [2] and were used with ≥98% purity. All other compounds were obtained from commercial sources, as follows: doxorubicin (DOX; Sigma-Aldrich, St. Louis, MO, USA), rapamycin (RAPA; Sigma-Aldrich, St. Louis, MO, USA), bafilomycin A1 (Selleck Chemicals, Houston, TX, USA), and Z-VAD-FMK (Cayman Chemical Company, Ann Arbor, MI, USA).

### 4.3. Cytotoxicity Analyses via the MTT Assay

Cytotoxic activity was evaluated by the transformation of 3-(4,5-dimethylthiazol-2-yl)-2,5-diphenyltetrazolium bromide (MTT; Thermo Fisher Scientific, San Jose, CA, USA) into formazan by viable cells [40]. Cells were plated in a 96-well plate (1 × 10^4^ cells/well) and, 24 h later, exposed to serial dilutions of the test compounds (from 0.00032 to 5.0 µM) for 24, 48, and 72 h. For each analysis, negative controls received 0.5% DMSO and positive control was delivered by doxorubicin (DOX; 0.05 to 10.0 µM). At the end of the incubation periods, the medium was replaced with fresh medium containing 0.5 mg/mL of MTT and supernatant was removed 3 h later. The plates were dried at 50 °C for 30 min and formazan was solubilized in DMSO (150 µL/well) for the absorbance measure at 570 nm on a plate reader (Thermo Fisher Scientific, San Jose, CA, USA). IC_50_ values (half-maximal inhibitory concentration) along with 95% CI (confidence intervals) were calculated by non-linear regression using GraphPad Prism 8.0 software (GraphPad Software, Inc., San Diego, CA, USA).

### 4.4. Selectivity Indices

Based on the respective IC_50_ values obtained through MTT assay (72 h), a selectivity index (SI) was determined for each tumor cell line treated with **SQ1** or **SQ2** in relation to a non-tumor cell by the following formula:SI = IC_50_ value from tumor cell line/IC_50_ from non-tumor cell line

### 4.5. Cell Viability Analyses through the Trypan Blue Exclusion Test

Fifty thousand (5 × 10^4^) cells in 1 mL of media were seeded in cell culture dishes and, 24 h later, treated with **SQ1** or **SQ2** (1.0, 2.5, and 5.0 µM), vehicle (DMSO) and doxorubicin (DOX, positive control) for 24 h. Cells were detached, stained with Trypan Blue solution 0.4% (1:10, *v*/*v*; Sigma-Aldrich, St. Louis, MO, USA) and counted in a Neubauer chamber under optical microscopy to determine concentrations of viable and non-viable cells.

### 4.6. Treatment Scheme

Twenty fifty thousand (2.5 × 10^5^) cells in 1 mL of media were seeded in cell culture dishes and, 24 h later, treated with **SQ1** (0.3 µM and 0.6 µM for SK-MEL-28 and 0.9 µM and 1.8 µM for SK-MEL-147) or **SQ2** (0.05 µM and 0.1 µM for SK-MEL-28 and 0.6 µM and 1.2 µM for SK-MEL-147) for 24 h. Doxorubicin (DOX; 1.4 µM for SK-MEL-28 and 0.8 µM for SK-MEL-147) was used as positive control and 0.5% DMSO as negative control. Concentrations were based on ½ IC_50_ values at 24 h and IC_50_ values at 24 h, respectively, from each cell line.

### 4.7. Clonogenic Assays

After the cells were submitted to the treatment scheme (Section 4.6), adherent cells were collected, plated at 1 × 10^3^ cell/plate on 35-mm plates and maintained in culture for 6 days in treatment-free medium. The medium was replaced every 2 days. Cells were then fixed and stained with a solution of 20% MeOH (Synth, Diadema, São Paulo, Brazil) and 0.5 g crystal violet (Sigma-Aldrich, St. Louis, MO, USA). Images were acquired using G:BOX Chemi XRQ (Syngene, Cambridge, UK) and quantified using the ImageJ program (ColonyArea; US National Institutes of Health, Bethesda, MD, USA).

### 4.8. Quantitative PCR Analyses

Treated cells (Section 4.6) had their total RNA extracted and purified using TRIzol reagent (Thermo Fisher Scientific, San Jose, CA, USA), in accordance with the manufacturer’s recommendations. For analysis of *DCD* expression, purified total RNA was quantified using an Eon Microplate Spectrophotometer (BioTek Instruments, Winooski, VT, USA). Quantitative PCR (qPCR) was conducted on a Rotor Gene 6000 real-time cycler (Corbett Life Science, Sydney, NSW, Australia) using GoTaq 1-Step RT-qPCR System (Promega, Madison, WI, EUA). The cycle conditions were as follows: 45 °C for 10 min, 95 °C for 15 min and 40 cycles of 95 °C for 15 s; 60 °C for 60 s, followed by the melt. Relative expression was calculated using the comparative CT method Delta-Delta-Ct (DDCt) [41]. The primer sequences and concentrations utilized (*DCD* and *RPLPO*) can be consulted in Appendix A.

For the gene panel, the total RNA from three independent samples was extracted using TRIzol reagent. The samples were quantified, and their quality assessed by NanoDrop 2000 One Microvolume UV-Vis Spectrophotometer (Thermo Fisher Scientific, San Jose, CA, USA). cDNA was synthesized from 1 µg RNA using a High-Capacity cDNA Reverse Transcription Kit (Thermo Fisher Scientific, San Jose, CA, USA). qPCR was performed using a QuantStudio 3 Real-Time PCR System in conjunction with a SybrGreen System (Thermo Fisher Scientific, San Jose, CA, USA). Relative expression was calculated using the comparative CT method Delta-Delta-Ct (DDCt) [38]. All primer sequences and concentrations utilized can be consulted in Appendix A.

### 4.9. Protein Expression Analyses by Western Blotting

Treated cells (Section 4.6) had their proteins extracted with the following solution: 10 mM ethylenediaminetetraacetic acid (EDTA), 10% triton, 2 mM phenylmethanesulfonyl fluoride (PMSF), 10 mM Tris base, 10 mM sodium pyrophosphate, 100 mM sodium fluoride, 2.5 mM sodium orthovanadate, and deionized water. The resulting extract was quantified for protein concentration and 30 µg of protein were subjected to SDS-PAGE and to Western blot analysis with the indicated antibodies: cleaved PARP-1 (#5625), PARP1 (#9542), cleaved caspase-3 (#9604), caspase-3 (#9662), SQSTM1/p62 (#5114), LC3B (#2775), and α-tubulin (#2144) from Cell Signaling Technology (Danvers, MA, USA). Images of membranes were made using Pierce ECL Western Blotting Substrate (Thermo Fisher Scientific, San Jose, CA, USA) and G:BOX Chemi XRQ (Syngene, Cambridge, UK). Band intensities were quantified using UN-SCAN-IT gel 6.1 software (Silk Scientific; Orem, UT, USA).

### 4.10. Statistical Analyses

Data were analyzed in GraphPad Prism 8.0 (GraphPad Software, Inc.) using one-way ANOVA followed by Dunnett’s test or Student’s *t*-test. Data are expressed in terms of mean and standard deviation (mean ± SD).

## 5. Conclusions

This study provides an in-depth evaluation of the biological activity of the natural product seriniquinone (**SQ1**) and synthetic analogue **SQ2** in melanoma cell lines. The resulting data provided new insights into the molecular MOA of this novel class of chemotherapeutic, as well as, revealing a further understanding of the mechanisms associated with induction of autophagy and apoptosis. Additionally, the cancer driver mutations (BRAF^V600E^ and NRAS^Q61R^) were found to not impact **SQ1** or **SQ2** cytotoxicity. The evidence herein suggests that a combination of **SQ1** or **SQ2** with autophagy inhibitors could be further explored to expand the efficacy of these agents and ideally enhance the targeting of BRAF mutated melanomas.

## Figures and Tables

**Figure 1 molecules-26-07362-f001:**
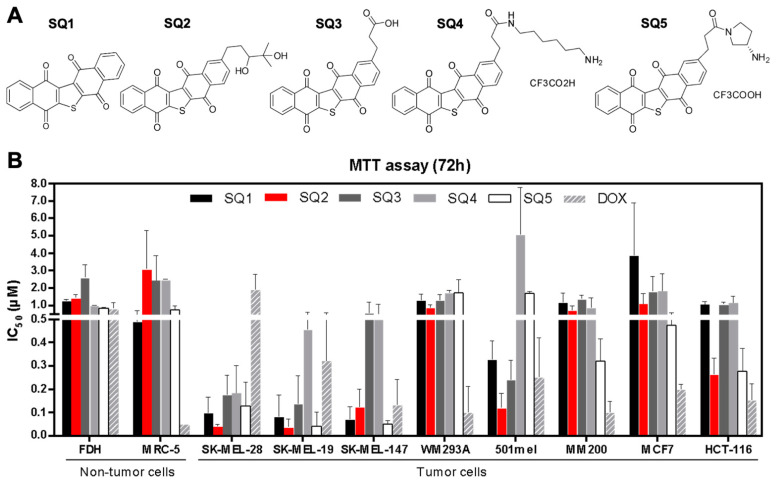
Seriniquinone (**SQ1**) and synthetic analogues (**SQ2**, **SQ3**, **SQ4**, and **SQ5**) are cytotoxic to cancer cell lines. (**A**) **SQ1**, the natural compound, was the prototype for designing the analogues. (**B**) All compounds were tested by 72-h treatment using the MTT assay in a panel of tumor and non-tumor cell lines, for which the IC_50_ values are expressed in µM (n = 3; mean ± SD). All experiments used DMSO (vehicle) and doxorubicin as negative and positive controls, respectively.

**Figure 2 molecules-26-07362-f002:**
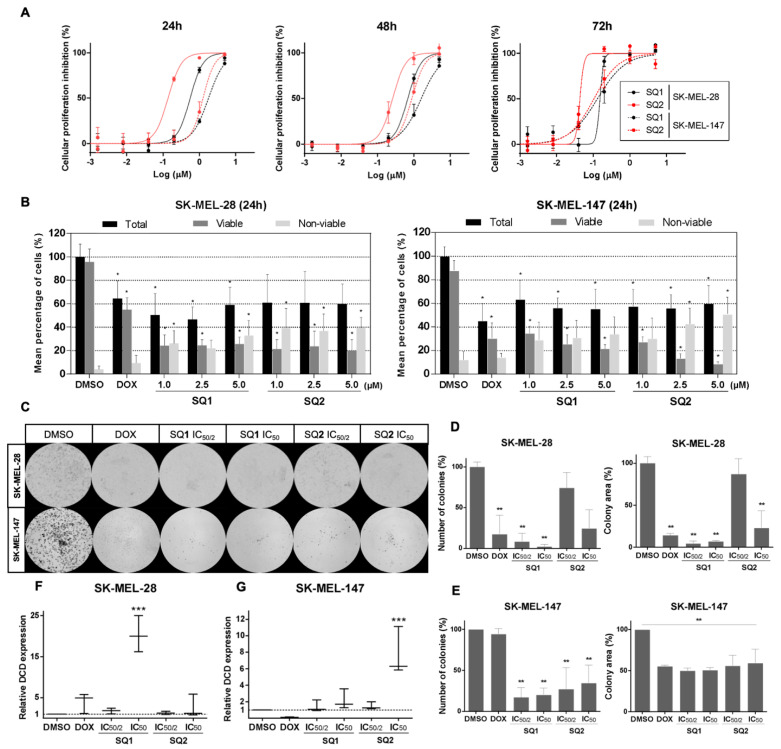
Characterization of **SQ1** and **SQ2** activity in BRAF and NRAS mutant cell lines. (**A**) Dose-response curves for **SQ1** (black) and **SQ2** (red) treatments of SK-MEL-28 (BRAF^V600E^ mutant; full line) and SK-MEL-147 (NRAS^Q61R^ mutant; dotted line) during 24-, 48-, and 72-h treatments. (**B**) **SQ1** and **SQ2** decreased the percentage of viable cells and increased non-viable cells. SK-MEL-28 and SK-MEL-147 were treated with 1% DMSO (negative control), 10 µM doxorubicin (DOX, positive control), **SQ1** and **SQ2** (1.0, 2.5, or 5.0 µM) for 24 h. With Trypan blue dye, cells were counted and viability was demonstrated as a percentage. (**C**) Representative images from the clonogenic assay. Cells were treated for 24 h with DMSO (negative control: 0.5%), doxorubicin (DOX; positive control: 1.4 µM for SK-MEL-28 or 0.8 µM for SK-MEL-147), **SQ1** (0.3 or 0.6 µM for SK-MEL-28 and 0.9 µM or 1.8 µM for SK-MEL-147) and **SQ2** (0.05 or 0.1 µM for SK-MEL-28 and 0.6 or 1.2 µM for SK-MEL-147). Viable cells formed colonies in 6 days in treatment-free medium. Quantifications (ImageJ program, ColonyArea) of number of colonies and colony areas (**D**) for SK-MEL-28 and (**E**) for SK-MEL-147. Relative expressions of *DCD* gene in (**F**) SK-MEL-28 and (**G**) SK-MEL-147 performed by qPCR (housekeeping gene: *RPLPO*). Cells were treated for 24 h with the same concentrations described for (**C**) and data are shown as box plots (minimum to maximum values). All the experiments were performed in biological triplicates (mean ± SD) and submitted to statistical analysis: ANOVA one way followed by Dunnett’s test (* *p* < 0.05, ** *p* < 0.002, *** *p* < 0.0005).

**Figure 3 molecules-26-07362-f003:**
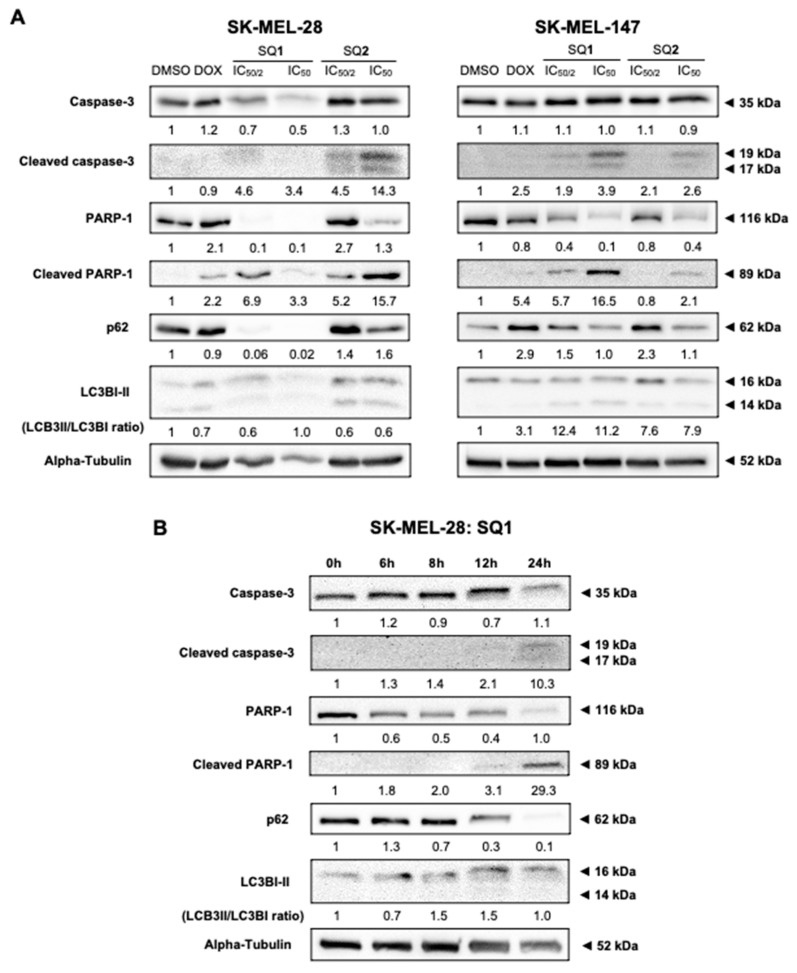
**SQ1** and **SQ2** induce cell death triggering autophagy and apoptosis. **SQ1** induce cell death marked by activation of autophagy and apoptosis. (**A**) SK-MEL-28 and SK-MEL-147 were treated for 24 h with DMSO (negative control: 0.2%), doxorubicin (DOX; positive control: 1.4 µM for SK-MEL-28 and 0.8 µM for SK-MEL-147), **SQ1** (0.3 and 0.6 µM for SK-MEL-28 and 0.89 and 1.79 µM for SK-MEL-147) and **SQ2** (0.05 and 0.1 µM for SK-MEL-28 and 0.63 and 1.26 µM for SK-MEL-147). (**B**) SK-MEL-28 was treated with the higher concentration of **SQ1** (0.6 µM) and proteins were extracted at times of 0, 6, 8, 12, and 24 h, demonstrating that autophagy and apoptosis processes occur concomitantly with initiation between 8 and 12 h. By Western blotting, proteins PARP-1, cleaved PARP-1, caspase 3, and cleaved caspase 3 were analyzed as apoptosis markers, while p62 and LC3B were probed for autophagy. Quantifications were carried out in the UN-SCAN-IT Gel 6.1 (Silk Scientific), normalized by comparison with α-tubulin and compared with the negative control of each experiment, showing the mean values (n = 3).

**Figure 4 molecules-26-07362-f004:**
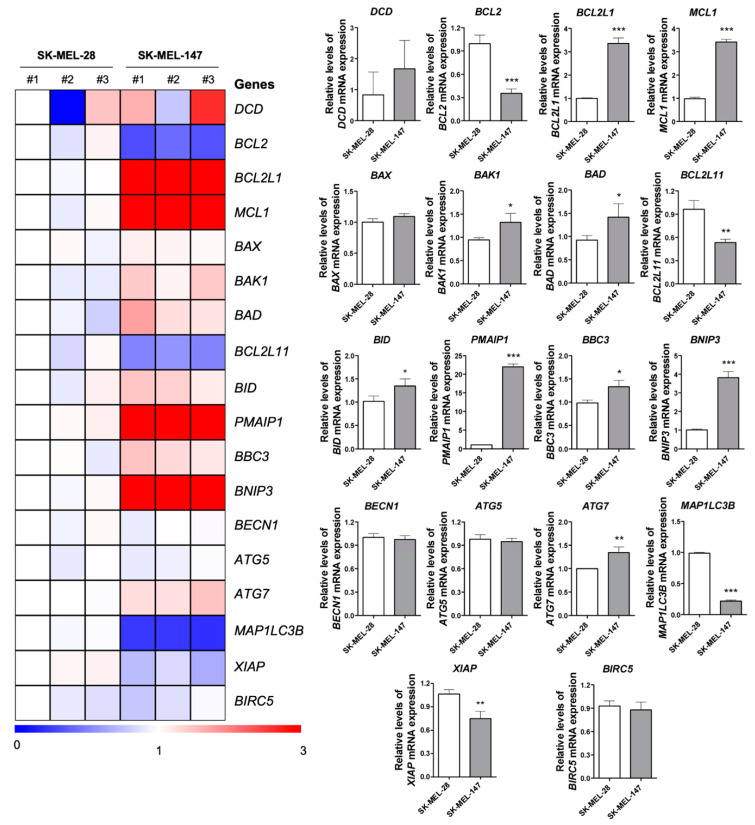
Basal expression of apoptosis and autophagy associated genes in BRAF^V600E^ and NRAS^Q61R^ melanoma cells. A heatmap depicting the levels of select apoptosis- and autophagy-related genes by qPCR analysis in three independent samples from SK-MEL-28 (BRAF^V600E^) and SK-MEL-147 (NRAS^Q61R^) cell lines. The data are presented as fold change of a sample from SK-MEL-28 cells. Downregulated and upregulated genes are illustrated in blue and red, respectively. Fold-changes of genes expression are presented in bar graphs (mean ± SD). The *p* values are indicated; * *p* < 0.05; ** *p* < 0.01; *** *p* < 0.001 as determined by a Student’s *t*-test.

**Figure 5 molecules-26-07362-f005:**
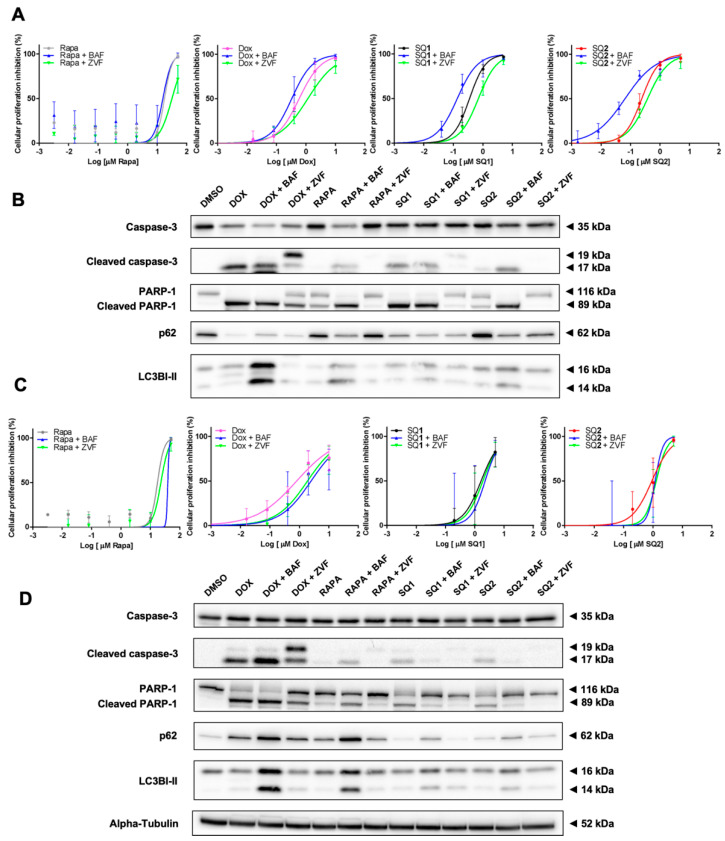
BRAF and NRAS respond differently under inhibition of the autophagy signaling. SK-MEL-28 (**A**,**B**) and SK-MEL-147 (**C**,**D**) cells were submitted to 1-h pre-treatments with 30 nM bafilomycin A1 (BAF) or 20 µM Z-VAD-FMK (ZVF). Dose-response curves (**A**–**C**) were calculated using the MTT assay with increasing concentrations of **SQ1** and **SQ2** following 24-h treatment (n = 3; mean ± SD). Protein expression (**B**–**D**) (PARP-1, cleaved PARP-1, caspase 3 and cleaved caspase 3, p62, and LC3B) was evaluated by Western blotting after 24 h of incubation with: DMSO (negative control: 0.2%), doxorubicin (DOX; positive control: 1.4 µM for SK-MEL-28 and 0.8 µM for SK-MEL-147), rapamycin (RAPA, 18 µM), **SQ1** (0.6 µM for SK-MEL-28 and 1.79 µM for SK-MEL-147), and **SQ2** (0.1 µM for SK-MEL-28 and 1.26 µM for SK-MEL-147). Quantifications were carried out in the UN-SCAN-IT Gel 6.1 (Silk Scientific), normalized by α-tubulin and compared with the negative control of each experiment (n = 3).

**Table 1 molecules-26-07362-t001:** Selectivity Index (SI) of **SQ1**, the synthetic analogues (**SQ2–SQ5**) and doxorubicin (DOX) for cancer (SK-MEL-19, SK-MEL-28, SK-MEL-147, WM293A, 501mel, MM200, MCF7, and HCT-116) versus normal fibroblast cells (MRC-5). Data is reported as the ratio of the respective 72-h IC_50_ value to that of MRC-5 cells. Highlighted in red, **SQ2** presented higher SI values than **SQ1**.

Selectivity Index (SI)								
	SK-MEL-28	SK-MEL-19	SK-MEL-147	WM293A	501mel	MM200	MCF7	HCT-116
**SQ1**	4.3	10.7	4.9	0.6	2.0	0.5	0.2	0.6
**SQ2**	39.3	39.3	15.7	2.1	14.3	2.1	2.2	6.8
**SQ3**	6.0	13.0	5.5	1.0	7.2	1.0	0.8	1.4
**SQ4**	12.9	7.7	6.1	1.4	-	2.1	2.5	2.4
**SQ5**	3.7	14.0	8.0	0.3	0.3	1.9	1.1	2.5
**DOX**	0.04	0.1	0.3	0.9	0.2	0.5	0.1	0.3

**Table 2 molecules-26-07362-t002:** Selectivity Index (SI) of **SQ1**, the synthetic analogues (**SQ2–SQ5**) and doxorubicin (DOX) for cancer (SK-MEL-19, SK-MEL-28, SK-MEL-147, WM293A, 501mel, MM200, MCF7, and HCT-116) versus normal primary fibroblast cells (FDH). Data was reported as the ratio of the 72-h IC_50_ value as compared to FDH cells. Highlighted in red, **SQ2** presented higher SI values than **SQ1**.

Selectivity Index (SI)								
	SK-MEL-28	SK-MEL-19	SK-MEL-147	WM293A	501mel	MM200	MCF7	HCT-116
**SQ1**	7.7	19.2	8.8	1.1	3.6	0.8	0.3	1.1
**SQ2**	29.3	29.3	11.7	1.6	10.6	1.6	1.6	5.1
**SQ3**	14.6	31.8	13.5	2.3	17.5	2.4	1.8	3.4
**SQ4**	5.3	3.2	2.5	0.6	-	0.9	1.0	1.0
**SQ5**	6.2	23.3	13.3	0.6	0.5	3.1	1.9	4.2
**DOX**	0.6	2.0	4.3	13.0	3.0	7.8	1.4	5.2

## Data Availability

The datasets used and/or analyzed during the current study are available from the corresponding author upon reasonable request.

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
