# Peer review of "Seriniquinones as Therapeutic Leads for Treatment of BRAF and NRAS Mutant Melanomas"

_molecules, 2021, doi:10.3390/molecules26237362_

Round 1

Reviewer 1 Report

The manuscript molecules-1459668, titled “Seriniquinones as therapeutic leads for treatment of BRAF and NRAS mutant melanomas”, by Hirata and colleagues, is an in vitro analysis of Seriniquinone molecules as potentially therapeutic drugs in melanoma, with possible implications for combination therapy in case of different BRAS and NRAS mutations. Overall, the study is well described in terms of cell phenotyping and range of tested lines.

Major points:

Even if a bit out of scope, this study could have been further improved by adding some in vivo tests.

Minor points

The whole manuscript should be checked to follow the International System of Units.

For instance, a space should be present between the numerical value and the unit symbol.

Results:

- Line 70: please define the IC50 acronym (inhibitory concentration, halving a response) at its first usage

- Figure 1: data dispersion should be displayed in the chart; doxorubicin (positive control) data should be displayed, for a comparison with SQ molecules. Are there significant differences among different molecules and cell lines?

The MTT assay should be briefly explained, for instance referring to it as “the cytotoxicity test MTT…”)

- Line 84: Selective indices “was” or “were” improved?

I would suggest to very briefly mention that the selectivity indexes were calculated through a comparison of drug activity on tumor versus non-tumor cells.

- Figure 2: charts should also display data dispersion

- Figure 2 legend and paragraph 2.3: human gene names should be written in (uppercase) italics

- Line 140: I would add a reference for the SQ1-dermcidin relationship

- Figure 3: a space should be present between numerical values and units of measure (e.g.: “35 kDa”)

- Paragraph 2.5: human gene names should be written in (uppercase) italics

- Figure 4: charts should display data dispersion

- Line 203: “ellicited” or “elicited”?

- Figure 5: Letterings in the figure and in the figure legend do not seem to correspond between each other. Is panel B referring to BRAF and panel D to NRAS?

If this is the case, there are some comments on panel B and panel D: could the authors comment on why alpha tubulin is highly changeable in panel B and so strikingly constant in panel D? Could the differences in the quality of the two sets of experiments have partially affected the comparisons between the two lines?

Does IB stand for Immuno-Blotting? Please specify.

- Table S3: the “Sequence” on top of the table should specify: 5’ to 3’

- Figure S1: a space should be present between numerical values and units of measure (eg: “35 kDa”)

Author Response

Answer to reviewers’ comments:

Reviewer #1

Major points:

Even if a bit out of scope, this study could have been further improved by adding some in vivo tests.

Answer: We agree that an in vivo evaluation would be an important proof-of-concept for seriniquinone anticancer activity, but unfortunately the compound has a very poor solubility. We tried to evaluate the in vivo activity through a collaboration with NCI, and the compound precipitated in the peritoneal cavity of treated animals. Even so, it was still possible to see antitumor activity. However, we cannot be sure of the concentration. Thus, the data are not publishable.

We’ve been addressing this issue using both medicinal chemistry (as showed in our previous paper published in ACS chemical Biology in 2020, https://doi.org/10.1021/acsmedchemlett.8b00391 - ref 2 in the present manuscript) and nanotechnology (as discussed in our previous paper published in Nanomedicine 2020 by Apolinario et al. 10.2217/nnm-2020-0290). So far, we were able to improve de solubility but still not enough to ensure in vivo assays. On the other hand, we developed some nanoformulation but up to now they were designed for topic application, and we are still working on other formation for systemic application. Thus, at this point, we are still not be able to add in vivo data.

Nonetheless, the present work showed interesting information of the mechanism of action of seriniquinones, discussing also the selectivity of these compounds to melanoma cells. These findings justify the continuation of the studies on seriniquinone anticancer activity.

Minor points

The whole manuscript should be checked to follow the International System of Units.

Answer: We checked all the units to follow International SI.

For instance, a space should be present between the numerical value and the unit symbol.

Answer: We carefully checked the presence of a space between the numerical value and the unit symbol throughout the manuscript.

Results:

- Line 70: please define the IC50 acronym (inhibitory concentration, halving a response) at its first usage

Answer: IC50 definition was included on page 2, line 73.

- Figure 1: data dispersion should be displayed in the chart; doxorubicin (positive control) data should be displayed, for a comparison with SQ molecules. Are there significant differences among different molecules and cell lines?

Answer: We changed the SEM to the Standard Deviation to better show the data dispersion. The information was included in figure legends and described in the material and methods section: “Data are expressed in terms of mean and standard deviation (mean ± SD)”.

We added doxorubicin data to Figure 1. We also included two brief comments on doxorubicin activity in comparison to SQs. “Doxorubicin was generally less toxic than seriniquinone and analogues considering melanoma cells, with the execption of WM293A and MM200 cells. Nonetheless, it was far more toxic to breast carcinoma MCF-7 cells and colon carcinoma HCT-116 cells, and also to the nontumor cells (MRC-5 and FDH)”. “Doxorubicin showed the lowest SI values for most tested cancer cells (Tables 1 and 2).”

The MTT assay should be briefly explained, for instance referring to it as “the cytotoxicity test MTT…”)

Answer: It was corrected.

- Line 84: Selective indices “was” or “were” improved?

Answer: It was corrected.

I would suggest to very briefly mention that the selectivity indexes were calculated through a comparison of drug activity on tumor versus non-tumor cells.

Answer: It was corrected.

- Figure 2: charts should also display data dispersion

Answer: We changed the SEM to the Standard deviation to better show the data dispersion. The information was included in figure legends and described in the material and methods section: “Data are expressed in terms of mean and standard deviation (mean ± SD)”.

- Figure 2 legend and paragraph 2.3: human gene names should be written in (uppercase) italics

Answer: It was corrected.

 - Line 140: I would add a reference for the SQ1-dermcidin relationship

Answer: It was corrected.

- Figure 3: a space should be present between numerical values and units of measure (e.g.: “35 kDa”)

Answer: It was corrected.

- Paragraph 2.5: human gene names should be written in (uppercase) italics

Answer: It was corrected.

- Figure 4: charts should display data dispersion

Answer: We changed the SEM to the Standard deviation to better show the data dispersion. The information were included in the figure legends and described in the material and methods section: “Data are expressed in terms of mean and standard deviation (mean ± SD)”.

- Line 203: “ellicited” or “elicited”?

Answer: It was corrected to ‘elicited’.

- Figure 5: Letterings in the figure and in the figure legend do not seem to correspond between each other. Is panel B referring to BRAF and panel D to NRAS?

Answer: It was corrected.

If this is the case, there are some comments on panel B and panel D: could the authors comment on why alpha tubulin is highly changeable in panel B and so strikingly constant in panel D? Could the differences in the quality of the two sets of experiments have partially affected the comparisons between the two lines?

Answer: Alpha tubulin is variable between the independent experiments, especially considering SQ toxicity to the cell. To avoid any interference, we always run alpha tubulin staining in the same membrane with other antibodies, then, we use these bands for normalization. The experiments were repeated multiple times, and the results were consistent considering independent experiments.

Does IB stand for Immuno-Blotting? Please specify.

Answer: We removed the IB mention, since it already mentioned that the analysis was performed by western blotting.

- Table S3: the “Sequence” on top of the table should specify: 5’ to 3’

Answer: It was corrected.

- Figure S1: a space should be present between numerical values and units of measure (eg: “35 kDa”)

Answer: It was corrected.

Reviewer 2 Report

The authors present a manuscript on the impact of seriniquinone treatment in melanoma.

The manuscript is of high quality and interesting to the field.

However some minor aspects need to be addressed before accepting the manuscript for publication

Most importantly, a limitations section needs to be included that clearly displays shortcomings and limitations of the manuscript. This section should be focused on future experiments that may close any gaps in the dataset/readouts and that further substantiates the impact of seriniquinone in glioblastoma.

Author Response

Reviewer #2

Most importantly, a limitations section needs to be included that clearly displays shortcomings and limitations of the manuscript. This section should be focused on future experiments that may close any gaps in the dataset/readouts and that further substantiates the impact of seriniquinone in glioblastoma.

Answer: We believe you meant melanoma, because so far we do not have data to seriniquinone in glioblastoma models. In the paper published by Trozz et al. (2014), seriniquinone was analyzed against the NCI-60 cell line panel, and the most sensitive cell lines were from melanoma and prostate. Nonetheless, considering your suggestion, we included a perspective on the main limitations of seriniquinone development as a chemotherapeutic drug to melanoma patients. We believe that the main issue is related to the limited solubility of compound, which, so far, does not allow effective in vivo evaluation. We added this discussion in the manuscript: “Additionally, to advance with seriniquinone into preclinical studies using animal models, it is necessary to overcome its poor water-solubility. As discussed by Apolinario et al [39], the development of nanotechnology-based formulations, as observed for other lipophilic natural compounds, including paclitaxel and doxorubicin, can be an effective strategy to enable in vivo studies with seriniquinone.”

Round 2

Reviewer 1 Report

No more comments.